# Study Protocol for a Randomised Controlled Trial Investigating the Effects of Maternal Prebiotic Fibre Dietary Supplementation from Mid-Pregnancy to Six Months’ Post-Partum on Child Allergic Disease Outcomes

**DOI:** 10.3390/nu14132753

**Published:** 2022-07-02

**Authors:** Debra J. Palmer, Jeffrey Keelan, Johan Garssen, Karen Simmer, Maria C. Jenmalm, Ravisha Srinivasjois, Desiree Silva, Susan L. Prescott

**Affiliations:** 1Telethon Kids Institute, The University of Western Australia, Nedlands, WA 6009, Australia; desiree.silva@telethonkids.org.au (D.S.); susan.prescott@telethonkids.org.au (S.L.P.); 2School of Medicine, The University of Western Australia, Crawley, WA 6009, Australia; jeff.keelan@uwa.edu.au (J.K.); karen.simmer@uwa.edu.au (K.S.); 3School of Biomedical Sciences, The University of Western Australia, Crawley, WA 6009, Australia; 4Division of Pharmacology, Utrecht Institute for Pharmaceutical Sciences, Faculty of Science, Utrecht University, 3584 CS Utrecht, The Netherlands; johan.garssen@danone.com; 5Nutricia Research, 3584 CT Utrecht, The Netherlands; 6Division of Inflammation and Infection, Department of Biomedical and Clinical Sciences, Faculty of Medicine and Health Sciences, Linköping University, 581 83 Linköping, Sweden; maria.jenmalm@liu.se; 7Joondalup Health Campus, Joondalup, WA 6027, Australia; srinivasjoisr@ramsayhealth.com.au; 8Department of Immunology and Dermatology, Perth Children’s Hospital, Nedlands, WA 6009, Australia; 9Nova Institute for Health, Baltimore, MD 21231, USA

**Keywords:** allergy prevention, allergen sensitisation, eczema, infancy, lactation, maternal supplementation, prebiotics, pregnancy

## Abstract

Infant allergy is the most common early manifestation of an increasing propensity for inflammation and immune dysregulation in modern environments. Refined low-fibre diets are a major risk for inflammatory diseases through adverse effects on the composition and function of gut microbiota. This has focused attention on the potential of prebiotic dietary fibres to favourably change gut microbiota, for local and systemic anti-inflammatory effects. In pregnancy, the immunomodulatory effects of prebiotics may also have benefits for the developing fetal immune system, and provide a potential dietary strategy to reduce the risk of allergic disease. Here, we present the study protocol for a double-blinded, randomised controlled trial investigating the effects of maternal prebiotics supplementation on child allergic disease outcomes. Eligible pregnant women have infants with a first-degree relative with a history of medically diagnosed allergic disease. Consented women are randomised to consume either prebiotics (galacto-oligosaccharides and fructo-oligosaccharides) or placebo (maltodextrin) powder daily from 18–20 weeks’ gestation to six months’ post-partum. The target sample size is 652 women. The primary outcome is infant medically diagnosed eczema; secondary outcomes include allergen sensitisation, food allergies and recurrent wheeze. Breast milk, stool and blood samples are collected at multiple timepoints for further analysis.

## 1. Introduction

Allergic diseases are now among the most common chronic diseases of childhood, with eczema/atopic dermatitis now affecting up to one-in-five infants and food allergy in one in ten infants in affluent societies [1]. Eczema/atopic dermatitis and food allergies commencing during infancy are the earliest manifestations of allergy, and reflect the vulnerability of the immune system to modern diet, lifestyle and environmental changes. Diet is a major determinant of gut microbiota composition and function [2,3], which in turn can be associated with inflammation and immune dysfunction [4]. Hence, the role of dietary composition in increasing commensal microbial gut diversity has emerged as one of the critical factors for beneficial immune health. 

Prebiotics are defined as substrates that are selectively utilized by host microorganisms conferring a health benefit [5]. Prebiotics are recognized as key immune-modulating nutrients, which selectively stimulate growth of beneficial gut microbiota—with prebiotic fermentation products, such as short-chain fatty acids (SCFA), that have direct local gut and systemic anti-inflammatory effects [5]. Microbial-derived SCFA are recognised as essential mediators in the communication between the intestinal microbiome mucus barrier and the immune system, and are considered as a likely pathway by which the gut microbiota influences immune health [6]. SCFA influence host immune responses through several key pathways [7] including: (1) promoting intestinal barrier integrity; (2) binding to ‘metabolite-sensing’ G-protein-coupled receptors such as GPR43, GPR41, and GPR109A, which promote gut homeostasis and the regulation of inflammatory responses [8,9,10]; and (3) inhibition of Histone Deacetylases which regulate rapamycin (mTOR)-S6K pathways required for T cell differentiation into effector and regulatory cells [11,12]. Along with the mTOR pathways, SCFA also enhance the activation of STAT3 (Signal transducer and activator of transcription 3), which is involved in the expression of cytokines (IL-10, IL-17, and IFN-γ) in T cells [12]. Collectively, these have the potential to alter the differential balance of Th1 versus Th2 immune differentiation, and as a consequence reduce the risk of allergic responses [11,12].

Two of the most studied prebiotics to date are galacto-oligosaccharides (GOS) and fructo-oligosaccharides (FOS). GOS are obtained either by synthesis or by extraction and hydrolysis of plant sources like pulses (α-GOS) or lactose (β-GOS) [13]. FOS are produced either by enzymatic synthesis or by hydrolysis of inulin from natural sources mainly from roots of chicory, artichoke, yacon, dahlia or agave [13]. Randomised controlled trials (RCTs) investigating the effects of GOS and FOS supplemented infant formula have found an increase in commensal *Bifidobacterium* growth [14,15,16,17], and reduced infant eczema incidence [18,19]. However, another RCT by Boyle et al. [20] found no reduction in infant eczema development, and a systematic review and meta-analysis which included 10 intervention trials (*n* = 4242 participants) using prebiotic supplemented infant formula found no overall reduction in eczema outcomes until four years of age (RR 0.75; 95% CI 0.56–1.01; I^2^ = 57%) [21]. Thus, the benefits of using infant prebiotic supplementation in the post-natal period as an allergy prevention strategy remain inconclusive. 

During pregnancy, there is close immunological interaction between the mother and her child providing enormous opportunities to influence fetal immune development [22,23,24]. Human fetal T cells are responsive to immune stimulation as early as 22 weeks’ gestation [25], and differences in immune function at birth in newborns who develop allergic disease indicates that ‘the scene is set’ to some extent by birth [26,27,28]. With evidence the fetal immune system is highly active and responsive to the mother and her diet, it is therefore logical to suggest that intervention with allergic disease prevention strategies during the in-utero period will have beneficial effects on later-life immune health. 

Animal models suggest that prebiotic fibre intake during pregnancy may influence immune programming in-utero and prevent allergic disease [10,29,30,31,32,33,34,35,36]. However, evidence in human studies on prebiotic fibre intake during pregnancy is still limited. Although two small human RCTs [37,38] have supplemented maternal diets with GOS and/or FOS from 25–26 weeks’ gestation, and found increased maternal faecal *Bifidobacterium* in the prebiotic compared to the placebo group, no child clinical allergic disease outcomes were assessed. One human observational study [10] has found an association between higher maternal dietary fibre intakes during pregnancy, increased maternal SCFA acetate levels, and reduced respiratory symptoms in the offspring (number of visits to the general practitioner for cough or wheeze). The observational findings from Thorburn et al. [10], along with the evidence from animal studies, support the need for well-designed RCTs to investigate the effect of maternal prebiotic intakes during pregnancy on child clinical allergic disease outcomes. Hence, the aim of this trial is to assess the effect of maternal prebiotic fibre dietary supplementation from 18–20 weeks’ gestation to six months’ post-partum to reduce the risk of child allergic disease development. The evidence generated from this trial will establish whether modulating the maternal diet and gut microbiota is a viable new earlier approach in the greatly needed strategies to prevent allergic diseases.

## 2. Trial Population and Methods

### 2.1. Trial Design and Ethical Approvals 

This is a two-arm (1:1 allocation), parallel-design, double-blinded, superiority randomised controlled trial. The trial is registered with the Australian New Zealand Clinical Trial Registry (ANZCTR): ACTRN12615001075572. The trial is known as the SYMBA Study. Ethical approval was obtained through the Joondalup Health Campus Human Research Ethics Committee (HREC Approval number 1451r); the University of Western Australia also granted reciprocal ethical approval for the trial (RA/4/1/8137). This trial is nested within the ORIGINS Project birth cohort [39] based at the Joondalup Health Campus, Western Australia, and the sponsoring institution is the Telethon Kids Institute, Nedlands, Western Australia.

### 2.2. Participant Eligibility

The inclusion criteria for this study are: (1) pregnant women < 21 weeks of gestation, whose infants have a first-degree relative (mother, father or sibling) with a history of medically diagnosed allergic disease (asthma, allergic rhinitis, eczema and/or food allergy); and (2) ability to provide informed consent. The gestational age is calculated from first trimester ultrasounds, or, if not available, the date of the last menstrual period. The exclusion criteria for this study are: (1) maternal smoking during pregnancy [40]; (2) maternal age < 18 years; (3) women already consuming prebiotic supplements (>twice per week); and (4) women who have a diagnosed cow’s milk allergy or lactose intolerance.

### 2.3. Recruitment

Women participating in the ORIGINS Project birth cohort [39] at the Joondalup Health Campus are given the trial participant information pamphlet. Pregnant women birthing at other hospitals in Western Australia are also informed about the trial by display of approved advertising material, including hard copy flyers and posters, as well as via digital media, which direct potential interested participants to contact our research staff. The women are then telephoned to follow-up if they are interested in participating, screened for eligibility, given full explanation in lay terms and any of their questions answered about the study. The participant is required to provide written informed consent and are given a copy of the signed Consent Form. Consent is voluntary and free from coercion, and women may withdraw their involvement in the trial at any time without prejudice and wherever possible the reason for withdrawal is recorded. Once eligibility is confirmed, basic demographic information (maternal age, postcode) is collected to comply with CONSORT guidelines [41].

### 2.4. Randomisation and Allocation to Treatment Groups

Once written consent has been obtained, each participant is randomised to one of two groups and assigned a unique uninformative five-digit study identification number (Study ID) according to a randomisation schedule via a secure web-based randomisation process. The Study ID number also corresponds to a matching Study ID labelled box of study powder tins to be provided to that participant. The computer-generated randomisation schedule is stratified, using randomly permuted, size-balanced blocks, by maternal allergy status (history of medically diagnosed asthma, allergic rhinitis, eczema and/or food allergy) and maternal pre-pregnancy body mass index (BMI < 25 or ≥25). The randomisation schedule is prepared by an independent statistician not involved with trial participants or data analysis. Randomisation codes are maintained by researchers working independently of the trial. Investigators and participants will not be unblinded until all the participating infants have completed their 12 month of age outcome assessments.

### 2.5. Interventions

The participating women are randomised to either a prebiotics intervention group or maltodextrin placebo control group:The prebiotics (intervention) group: women are asked to consume 14.2 g per day of prebiotic powder (Galacto-oligosaccharides (GOS) and Fructo-oligosaccharides (FOS) ratio 9:1) from <21 weeks gestation until 6 months’ postnatal infant age;The maltodextrin (control) group: women are asked to consume 8.7 g per day of maltodextrin powder from <21 weeks gestation until 6 months’ postnatal infant age.

The daily dose amounts of each study powder were designed to ensure matching energy content of 143 KJ per day. Both study powders are mixed with food or beverage and consumed orally by the participating women once per day. During the randomisation appointment, the research staff demonstrate how to use and level of the provided study powder scoop in the tin accurately to measure out the daily study powder amount, using an example scoop and cow’s milk powder (not the actual allocated powder scoop). An information leaflet with study powder mixing instructions is provided to each participant at the appointment. The participants are instructed not to consume any other prebiotic products during the trial intervention period

### 2.6. Intervention Duration

The intervention period for this trial is from study entry (18–20 weeks gestation) until 6 months of lactation. We chose a pragmatic and realistic approach to the dietary intervention. This was carefully considered to replicate the likely ‘real life’ scenario where a maternal healthy ‘high prebiotic fibre diet’ would not cease at birth of the baby, but continue during lactation. Furthermore, a combined pre- and postnatal strategy may also have enhanced allergy preventive effects (as discussed in [22]).

### 2.7. Blinding

Participants, outcome assessors, research staff and data analysts are blinded to the randomisation group allocation. To maintain the blind, both the intervention (prebiotic) and control placebo (maltodextrin) powders are packaged in identical tins. Both study powders are similar in taste, texture and colour. The difference in daily dose weights are accommodated by the study powders having different size scoops. The scoops are concealed within the tins to ensure blinding. All study powders are manufactured, packaged, and labelled by the independent company, Danone, Netherlands, according to Good Manufacturing Procedures. The trial is an investigator-initiated project and Danone had no other involvement other than the provision of the study products. The study powder supplier packaged and coded both trial study powders with the unique five-digit Study ID number. This study powder supplier is not involved in the assessment process. Study powders are securely stored at Joondalup Health Campus and clearly labelled for research purposes.

### 2.8. Monitoring Adherence to Study Powder Ingestion

During the intervention period the research team maintain regular (in-person or telephone) monthly contact with participants to monitor and encourage study powder consumption adherence, as well as to answer any questions as they arise. If the participant stops taking the study powder, she is asked to provide a reason for ceasing study powder ingestion. The participating women return their unused powder tins for weighing at the end of the intervention period when their infant is six months of age.

### 2.9. Trial Outcomes

The primary outcome for this trial is infant medically diagnosed eczema by 1 year of age. 

The secondary clinical outcomes (all participants) are: Infant medically diagnosed IgE-mediated food allergies by 1 year of age.Infant medically diagnosed recurrent wheeze by 1 year of age.Infant allergic sensitisation to food and environmental allergens at 1 year of age.Maternal and infant skin barrier permeability at 36 weeks (maternal only), 3–4 and 6 months post-natal, and at 1 year of age (infant only).Maternal weight gain during the pregnancy and lactation intervention period, gestational length, and pregnancy complications (including gestational diabetes mellitus, pre-eclampsia and gestational hypertension).Infant birth anthropometrics and body composition.Infant weight gain and growth during infancy.The secondary laboratory outcomes are:Maternal immune responses at 20, 28 and 36 weeks of gestation.Maternal gut microflora colonisation patterns and stool short-chain fatty acid levels and composition during the pregnancy and lactation intervention period.Infant gut microflora colonisation patterns and stool short-chain fatty acid profiles at 2, 3–4, 6 and 12 months of age.Infant immune responses at birth, 3–4, 6 and 12 months of age.Breast milk composition and immune components at 2, 3–4, 6 and 12 months of age.

### 2.10. Clinical Outcome Definitions and Clinical Assessment Methods

#### 2.10.1. Infant Medically Diagnosed Eczema

Infant medically diagnosed eczema is defined by typical eczema skin lesions [42] clinically observed by a medical practitioner. The extent and severity of any active eczema at the in-person infant review appointments will be scored according to the standardized SCORAD severity index [43].

#### 2.10.2. Infant Medically Diagnosed Recurrent Wheeze

Infant medically diagnosed recurrent wheeze is defined as more than one separate episode of wheeze symptoms clinically observed by a medical practitioner.

#### 2.10.3. Infant Allergen Sensitisation

Infant allergen sensitisation is defined as a positive skin prick test (with mean weal diameter ≥ 3 mm above the control weal size) to at least one food and/or environmental allergen at 1 year of age. The participating infants are skin prick tested to egg, cow’s milk, wheat, fish (tuna), peanut, cashew nut, ryegrass pollen, cat and house dust mite (Dermatophagoides pteronyssinus), with histamine and control solutions, using commercial extracts, and in accordance with standard clinical methods.

#### 2.10.4. Infant IgE-Mediated Food Allergy

Infant IgE-mediated food allergy is defined as a history of immediate IgE-mediated symptoms (within 60 min of food ingestion) including angioedema, urticaria, cough, wheeze, stridor, vomiting, diarrhoea and/or cardiovascular symptoms; and specific food allergen sensitisation to the same food detected by a positive skin prick test (as described above).

#### 2.10.5. Skin Barrier Permeability

Maternal and infant skin barrier permeability is assessed by measuring trans epidermal water loss (TEWL), a measure of which determines to the skin’s ability to retain moisture, measurements are conducted using the Biox Aquaflux AF200 (Biox Systems Ltd., London, UK).

#### 2.10.6. Body Composition

Infant birth body composition is assessed by using the through air-displacement plethysmography PEA POD body composition system (Cosmed, Rome, Italy) up to 96 h after birth to measure infant fat mass, fat-free mass, percentage body fat and percentage fat-free mass.

### 2.11. Laboratory Outcomes Assessment Methods

Biological samples (blood, stool and breast milk) analyses will be done on pre-determined subsets of the participants to determine the effects of the intervention. The use of longitudinal biological sample collections, and assessing changes from the 20 weeks’ gestation baseline, will reduce potential influences of inter-women variations in chronic health status, diet, lifestyle and environment. The results of these laboratory outcome assessments are planned to be published separately to the primary outcome clinical findings paper. 

#### 2.11.1. Blood Samples

Blood samples (maternal, cord blood and infant) are being analysed for innate and adaptive immune responses. These include maternal serum inflammatory biomarkers (especially IL-1β, IL-6 and TNF-α); cord blood and infant cytokines responses, including IL-6, TNFα, IL-1β, IL-12 and IL-10); effector T cell responses to mitogens and allergens, and Th2 (IL-13, IL-5), Th1 (IFNγ) and IL-10 responses in the supernatants; and regulatory T cell responses, including the numbers and function of CD4(+)C25(+)CD127(low/−) cells, and expression of regulatory marker FOXP3+. 

#### 2.11.2. Stool Samples

Stool samples (both maternal and infant) are being analysed for gut microflora colonisation patterns and stool short-chain fatty acid profiles. Our prime approach is based on extraction of DNA, amplification using 16S primers, and sequencing using the Illumina MiSeq platform. We are assessing qualitative and quantitative information on bacterial species, microbial diversity and ordination. Individual stool sample (both maternal and infant) short chain fatty acid (SCFA) concentrations of acetic, propionic, butyric, isobutyric, valeric, and isovaleric acid are also being analysed, along with determining the relative proportions of the sum of all acids quantified.

#### 2.11.3. Breast Milk Samples

Breast milk samples are being analysed for the measurement of immunoglobulins (including secretory IgA, allergen-specific IgA and IgG4); cytokines/chemokines (including IL-1beta, IL-4, IL-5, IL-6, IL-10, IL-13, IL-27, IL-33, and TNF alpha); growth factors (including epidermal growth factor and insulin like growth factor-I); human milk microbiome; human milk oligosaccharide profiles; and macro- and micro-nutrient composition. 

### 2.12. Study Timeline

Figure 1 summarises the participants schedule of appointments and biological sample collections. Before 21 weeks’ gestation, informed consent including a screening form checklist is completed. At 18–20 weeks’ gestation baseline data are recorded, including basic demographic information; maternal, paternal and any sibling history of allergic disease; maternal and paternal ethnicity and education; and any prior prebiotics or probiotics supplements use. Maternal weight and height are measured, and pre-pregnancy weight recorded. Baseline blood and stool samples are collected. 

Each month during the intervention period, from 18–20 weeks’ gestation until 6 months’ post-partum, the participating women are asked about their compliance with the study powder ingestion, any adverse gastrointestinal symptoms, stool frequency and consistency, any use of other prebiotics, probiotics or antibiotics, and any hospitalisations for >24 h duration. Maternal in-person appointments in pregnancy also occur at 28 and 36 weeks’ gestation, when weight and TEWL (36 weeks only) are measured, and blood and stool samples are collected.

After birth, data on infant sex, birth weight, length, head circumference, and mode of delivery details are collected. Infants also have a body composition measurement up to 96 h after birth. A sample of cord blood is collected where possible at delivery. 

Postnatal telephone calls occur at 1, 2 and 5 months of age. Data are also collected to prospectively capture information on infant feeding and to document any infant allergy symptoms. At the 1 month phone call, participants are reminded to collect a breast milk sample, as well as a maternal and infant stool sample when the infant is 2 months of age. These are frozen in the home freezer until they are brought in at the 3–4 month visit.

Postnatal appointments occur at 3–4 and 6 months of age, when maternal weight and TEWL is measured, and infant TEWL and anthropometric measurements (length, weight, head circumference) are taken. Data collection also includes infant feeding (including solid foods introduction), use of any probiotics or antibiotics, hospitalisations for >24 h duration, and history of wheeze and/or eczema symptoms. The infant is examined for clinical signs of allergic disease (ie eczema), and SCORAD [43] (severity and extent of eczema) assessments performed if relevant. At these visits the following samples are also collected: infant blood, maternal blood (3–4 months only), infant and maternal stool and breast milk. 

At 1 year of age (between 12–18 months of age), a child follow-up appointment will collect data on infant feeding (including solid foods introduction), use of antibiotics, hospitalisations (>24 h duration), history of wheeze and/or eczema symptoms, examination for clinical signs of allergic disease (i.e., eczema), and SCORAD (severity and extent of eczema) if relevant. Child TEWL and anthropometric (length, weight, head circumference) measurements are taken. Child blood, stool and breast milk samples are collected. The child also has allergy skin prick testing to determine allergen sensitisation status as described above.

### 2.13. Serious Adverse Event Monitoring

A blinded independent Serious Adverse Event (SAE) Committee consists of an obstetrician, an allergist/immunologist and a neonatologist/paediatrician. This committee meets as required to review any deaths, admissions to intensive care hospital care or other serious adverse reactions. The primary role of the SAE Committee is to review all SAEs to determine whether there is any likelihood that involvement in the trial could have contributed to the event. The cause of the event is determined from the autopsy results or other hospital summary of the event by the relevant medical personnel. In case of any potential safety concerns observed by the SAE Committee, these are communicated to the trial Chief Investigators (DJP and SLP) who also inform the governing Ethics Committee.

### 2.14. Data Collection and Management

Trial monitoring to ensure compliance with good clinical practice and the study protocol is conducted by a Chief Investigator (DJP) every six months, or as required, to ensure the integrity of the trial. Data are collected by trained research personnel and entered directly into an electronic case-report form with password protection and defined user-level access. Research Electronic Data Capture (REDCap) is used to facilitate trial management and data collection. A record of all women successfully screened for eligibility and consented is recorded in real time. Once consented and randomised, REDCap automatically calculates study milestones for each participant. This information is readily available for research staff to enable scheduling of appointments and phone calls. The electronic case report form has inbuilt data entry validity checks to ensure immediate resolution of data queries. Summary reports including screening data, enrolment, withdrawals, appointment attendance and study completion are generated from REDCap for regular Chief Investigator review of trial progress. Electronic data are stored on secure servers at the Telethon Kids Institute with access only granted to authorised study personnel and released only to persons authorised to receive those data. All data collected is treated with confidence.

### 2.15. Sample Size

Based on infant medically diagnosed eczema by 12 months of age as the primary outcome, with an expected background rate of 30% in this ‘at risk’ population due to family history of allergic disease [44,45], 293 participants per group will be required to achieve a relative reduction of 33% (to a rate of 20%) in the intervention group (two-tailed alpha = 0.05, 80% power). Maternal probiotics supplementation [46]) has been shown to significantly reduce infant eczema in a smaller ‘at risk’ population. Allowing for a 10% dropout, a total sample size of 652 (326 per group) allows for an adequate sample size for this trial.

### 2.16. Statistical Analysis

Analyses will be performed on an intention-to-treat basis (i.e., all randomised women analysed as randomised). For the primary outcome, the proportion of infants with medically diagnosed eczema by 12 months of age will be compared between groups using log binomial regression. Adjustment will be made for variables used to stratify the randomisation, with the difference between groups expressed as an adjusted relative risk with a confidence interval and two-sided *p*-value. Differences in secondary clinical outcomes will be examined in the same way. A sensitivity per-protocol analysis of the primary outcome will be undertaken in women that consumed at least 75% of their allocated study powder (based on return weights) during the entire intervention period (<21 weeks gestation to 6 months post-natal). Another sensitivity per-protocol analysis of the primary outcome will be undertaken in women that consumed their allocated study powder on at least 75% of the days between randomisation and 36 weeks of gestation (last clinic visit prior to birth).

### 2.17. Current Trial Status

The first participant was randomised on 29 June 2016 and recruitment was completed on 18 November 2021, with 652 women randomised into the trial and our full sample size aim achieved. All infant 1 year of age primary outcome assessments are expected to be completed by June 2023.

## Figures and Tables

**Figure 1 nutrients-14-02753-f001:**
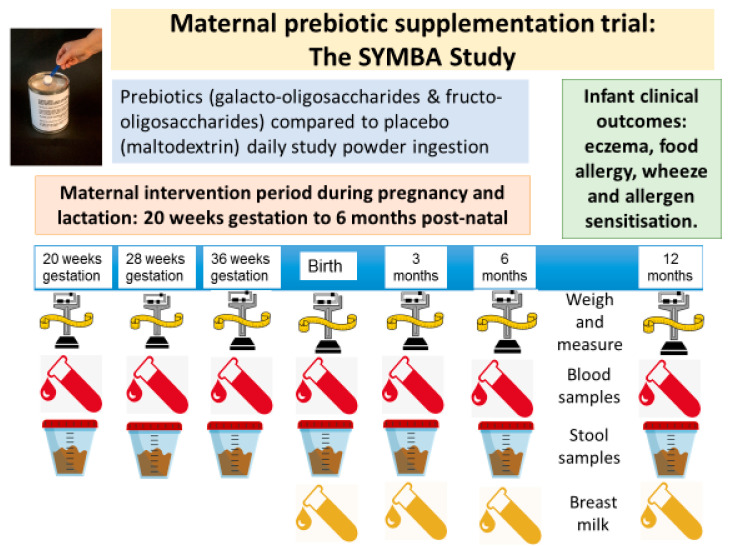
**Figure 1** summarizes the participants (maternal and child) schedule of appointments and biological sample collections. The maternal intervention period is from 18–20 weeks’ gestation until 6 months’ post-partum, when the participating women are asked to consume their allocated study powder daily. Child assessments occur at 3, 6 and 12 months of age.

## Data Availability

Once the primary outcome of the trial is published, the SYMBA Study data will be available for data sharing. Data sharing requests will need approval by the SYMBA Study Investigator Team. Please send requests to the corresponding author (debbie.palmer@telethonkids.org.au). The Australian National Health and Medical Research Council (NHMRC) supports the sharing of outputs from NHMRC funded research including publications and data. All recipients of NHMRC grants must therefore comply with all elements of the NHMRC Open Access Policy (15 January 2018).

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
