# Peer review of "Study Protocol for a Randomised Controlled Trial Investigating the Effects of Maternal Prebiotic Fibre Dietary Supplementation from Mid-Pregnancy to Six Months’ Post-Partum on Child Allergic Disease Outcomes"

_nutrients, 2022, doi:10.3390/nu14132753_

Round 1

Reviewer 1 Report

In this manuscript, the author developed a protocol for a double-blind, randomized controlled trial in investigating the effect of maternal prebiotic supplementation on allergic disease outcomes in children. This protocol is rigorous, thorough and detailed, and will provide a reliable experimental design reference for future researchers. However, the current manuscript still suffers from some flaws, which must be comprehensively discussed and addressed by the authors before the manuscript possibly suitable for publication, as indicated below:

1. In line 123, the author calculated the gestational age by the date of the last menstrual period. However, there probably be individual differences for those with irregular periods, the author should provide the specific method to calculate the gestational age.

2. Chronic diseases affect host physiological markers and gut microbes, as well as short-chain fatty acids. The authors collected blood samples and stool samples to detect inflammatory biomarkers and gut microbiota. Have the authors considered the impact of maternal long-term chronic diseases?

3. In line 163, the daily dose amounts of each study powder were designed to ensure matching energy content of 143 KJ per day. Why the author chose the same energy content but not the same weight of study powder? If the authors considered the same energy intake, the diet of maternal should be recorded too.

4. In line 124, the authors should provide references for the reasons for excluding pregnant women who smoked during pregnancy. In addition, should maternal who consume narcotics and psychotropic drugs be excluded?

5. In line 198, are women who did not consume enough powder considered to be excluded or not?

Reviewer 2 Report

Very good study design.

I believe that  everything (introduction writing, method design, and discussion) is fine.

Author Response

We thank Reviewer 2 for their positive comments.